# Internalizing symptoms and family functioning predict adolescent depressive symptoms during COVID-19: A longitudinal study in a community sample

Stefania V. Vacaru[1]*, Roseriet Beijers[1,2], Carolina de Weerth[1]

1 Donders Institute for Brain, Cognition & Behavior, Radboud University Medical Center, Nijmegen, Netherlands, 2 Radboud University, Nijmegen, The Netherlands

* stefania.vacaru@radboudumc.nl

## Abstract

### Background

The COVID-19 pandemic and lockdown pose a threat for adolescentsâ mental health, especially for those with an earlier vulnerability. Accordingly, these adolescents may need increased support from family and friends. This study investigated whether family functioning and peer connectedness protects adolescents with earlier internalizing or externalizing symptoms from increased depressive symptoms during the first Dutch COVID-19 lockdown in a low-risk community sample.

### Methods

This sample comprised 115 adolescents ($M_{age}$ = 13.06; 44% girls) and their parents ($N$ = 111) and is part of an ongoing prospective study on child development. Internalizing and externalizing symptoms were self-reported a year before the COVID-19 lockdown. In an online survey during the first Dutch lockdown (April-May 2020), adolescents reported depressive symptoms and perceived peer connectedness, and parents reported family functioning.

### Results

Twenty-four percent of adolescents reported clinically relevant symptoms of depression during the first COVID-19 lockdown. Depressive symptoms were significantly predicted by earlier internalizing, but not externalizing symptoms. Furthermore, higher quality of family functioning, but not peer connectedness, predicted fewer adolescent depressive symptoms. Family functioning and peer connectedness did not moderate the link between pre-existing internalizing symptoms and later depressive symptoms.

### Conclusions

In a low-risk community sample, one-in-four adolescents reported clinically relevant depressive symptoms at the first COVID-19 lockdown. Higher earlier internalizing symptoms and

**Data Availability Statement:** The dataset on which these analyses are conducted belong to the historical longitudinal BIBO (Basale Invloeden op de Baby Ontwikkeling) study, which started in

2006. In accordance with the informed consent approved by the Ethical Committee of the Faculty of Social Sciences, Radboud University, Nijmegen (ECG300107/SW2017-1303-497/SW2017-1303-498) and signed by the participants, data cannot be shared in public repositories. However, access to the data can be requested to the data access manager Irene van Kroonenburg (irene.vankroonenburg@radboudumc.nl).

**Funding:** The BIBO study was supported by a Netherlands Organization for Scientific Research VIDI grant (575-25-009, to CdW), VICI grant (016.185.038, to CdW), VENI grant (016.195.197, to RB), an Early Career Award of the Royal Netherlands Academy of Arts and Sciences (to RB). The funders had no role in study design, data collection and analysis, decision to publish, or preparation of the manuscript.

**Competing interests:** The authors have declared that no competing interests exist.

lower quality of family functioning increased risks. These results indicate that even in low-risk samples, a substantial group of adolescents and their families are vulnerable during times of crisis.

## Introduction

Adolescence is a sensitive period that sees important biological and social changes [1]. These changes may put adolescents at risk for mental health problems, with increased depression rates in teens [2], particularly when experiencing heightened stress (i.e., natural disasters; [3]). After the outbreak of COVID-19, governments worldwide implemented unprecedented lock-down measures, including social distancing, and closing of schools, sports, and entertainment venues [4]. As social environments play an important role for youthsâ social development and well-being [5,6], the COVID-19 outbreak and lockdown may have had a psychological impact on adolescents. Indeed, up to 41.7% of youth were found to suffer from mental health problems during the lockdown (for a meta-analysis see [7]). However, not all adolescents will be impacted by the lockdown in the same way. While some adolescents may have a heightened vulnerability due to pre-existing conditions such as mental health issues [8], others may have protective factors that buffer against negative environmental influences [9]. Consequently, knowledge of the interrelations between prior vulnerabilities and possible counteracting protective factors is needed for prevention and timely intervention during the ongoing and future crises.

Experiencing a stressful life event with pre-existing psychological symptoms and behavior problems may put adolescents at higher risk for negative consequences [10]. Internalizing symptoms include emotional and peer problems, such as anxiety, depression and social withdrawal, whereas externalizing symptoms are characterized by conduct problems, such as anger, aggression, and hyperactivity [11]. Adolescents with such behavioral symptoms may be additionally challenged during a lockdown because of a drastic disruption to their (social) environments, fewer possibilities to engage in support activities, or limited access to mental healthcare due to a high demand on an overwhelmed system in early 2020 [12]. Schools also often provide support and mental health programs, and their closure may hold important implications for childrenâ s wellbeing [13]. Previous work showed a deterioration in youthâ s mental health during the current and past health-related crises [14,15]. Contrarily, others have found a reduction in adolescentsâ symptoms across several mental health domains one month into lockdown as compared to one month prior [16]. These mixed findings suggest that such crises have an impact on adolescentsâ psychological wellbeing, yet it remains unclear whether it is for the better or for the worse, and for whom. The first goal of this study is to determine whether adolescents from a community sample experienced depressive symptoms during the first Dutch COVID-19 lockdown, and whether adolescents with pre-existing internalizing/externalizing symptoms were especially vulnerable.

While pre-existent psychological symptoms may increase an adolescentâ s vulnerability for developing depressive symptoms during times of crisis, protective factors may help prevent problems. Being housebound during the COVID-19 lockdown, the family environment may play a fundamental role for adolescent mental wellbeing. COVID-19 regulations regarding working from home and online education have resulted in adolescents spending most of their day under the same roof with their mothers, fathers, siblings and/or other potential family members. While prior work indicated that parent-child relationships occupy a central role in

buffering childrenâ s stress [17,18], in particular parental structure (e.g. organization, consistency and routine) and parental responsiveness, indices of the parent-child relationship (e.g. how individuals behave or how sub-systems within the system behave) do not reflect the whole family functioning as a system [19]. Indeed the overall functioning of a family and changes within the family may for instance relate differently to childrenâ s psychological wellbeing [20]. According to the family systems theory, the family environment is conceptualized as a dynamic system of a multitude of dimensions, such as communication, affect, autonomy, which together are fundamental for a childâ s growth and physical and psychological wellbeing [21]. While reports during the COVID-19 lockdown indicated that poor or deteriorating parent-child relationships contribute to worse childrenâ s mental health problems [16,22,23], it still remains to be determined whether overall family functioning buffers vulnerable adolescentsâ risk for developing depressive symptoms.

Besides the family environment, the social network constitutes a powerful buffer against stress [9]. Adolescence is marked by profound identity and social transformations, as adolescents grow more independent from their parents and start identifying more with peers [24]. Positive and supportive peer relationships were shown to mitigate the negative effects of adversities [25]. Evidence from disaster studies shows that social support disruptions following a disaster may affect youthsâ ability to cope with the stressor and lead to mental health sequelae [26]. The COVID-19 lockdown and the associated social distancing regulations has similarly disrupted social dynamics, leading to feelings of loneliness (for a systematic review see: [27]). Nonetheless, adolescents may find alternative ways of connecting with their peers, such as at 1.5m distance in real life, or digitally, via online social media. This may render adolescents more resilient to detrimental psychological outcomes. A recent study showed that a larger social network size predicts less distress during the pandemic [28], suggesting the importance of social connectedness during this major crisis. However, adolescentsâ social network size may not always reflect subjective feelings of how connected they feel with their peers [29]. The second goal of this study is to determine whether family functioning quality and feelings of peer connectedness during the first Dutch lockdown buffer potential relations between prior internalizing/externalizing symptoms and depressive symptoms.

This is an ongoing longitudinal study on a low-risk community sample. Internalizing/externalizing symptoms were assessed via self-report at age 12 in 2019, whereas family functioning was reported by parents, and peer connectedness and depression symptoms were self-reported at age 13 in an online questionnaire in the first Dutch lockdown (April/May 2020). In line with previous evidence that prior childhood psychological vulnerability (e.g. internalizing and externalizing symptoms [30]) is linked to depression later in life [31], especially during times of crises [3], we hypothesized that having either higher internalizing or externalizing symptoms would be linked to higher depressive symptoms during the first COVID-19 lockdown, but not when the quality of family functioning and connectedness with peers were high.

## Methods

### Participants

This sample is part of a larger ongoing prospective study that follows a community sample of mothers and their offspring since pregnancy (N = 193, [32]). The data used in this study belongs to two separate waves: the year prior the pandemic (2019; *Mage* = 12.66 years; *SDage* = 0.30) and during the pandemic (April/May 2020; *Mage* = 13.61 years; *SDage* = 0.40). One hundred fifteen adolescents (51 girls) and their mothers (N = 111) participated in an online survey that assessed their experiences with the first COVID-19 lockdown and regulations between April-May 2020, during the most stringent lockdown restrictions in The

Netherlands and when the schools were closed. All participants were born between January 2006 and July 2007, and were of Dutch ethnicity. All parents provided written informed consent for their own and their children participation in the study. The BIBO study (Dutch acronym for Basal Influences on Baby Development) was approved by the ethics committee from the Ethical Committee of the Faculty of Social Sciences, Radboud University, Nijmegen (ECG300107/SW2017-1303-497/SW2017-1303-498) and was performed following the Declaration of Helsinki principles.

## Measures

**Depressive symptoms.** During the first lockdown, the adolescents filled in *The Center for Epidemiological Studies Depression Scale for Children* (CES-DC; [33]), which was validated in The Netherlands and internationally [34,35]. It comprises 20 items on 4-point Likert scales from 0 = never to 4 = always, assessing symptoms in the past week (e.g., â I felt depressedâ ). Total scores range between 0â 60, with higher scores indicating more depressive complaints. A cut-off score of â¥16 discriminates clinically relevant symptoms of depression. The internal consistency was good (Cronbach Î± of .85).

**Internalizing/Externalizing symptoms.** The adolescents filled in the *Strengths and Difficulties Questionnaire* (SDQ; [36]) in the year preceding the lockdown. This is a well-established self-report questionnaire comprising 25 items on a 3-point Likert scale, with 0 = not true, 1 = somewhat true and 2 = certainly true. Higher scores indicate higher levels of symptoms. The raw scores of the items are summed and divided into five subscales: emotional problems (EP), conduct problems (CP), hyperactivity/inattention (Hyp), peer problems (PP) and prosocial behaviors (PB). The first four subscales are further grouped into two higher dimensions: internalizing (EP+PP) and externalizing symptoms (CP+Hyp). A cut-off score of â¥7 for internalizing symptoms and â¥8 for externalizing symptoms is used for Dutch low-risk 12-to-18-year-olds [36]. Internal consistency was moderate to high, with Cronbach Î± coefficients of .70 for externalizing and .71 for internalizing subscales.

**Family functioning.** Examined during the lockdown via parent-report with the *McMaster Family Assessment Device* (FAD; [19]). This instrument includes 60 items on a 4-point Likert scale from 1 = strongly agree to 4 = strongly disagree, which reflect different dimensions of the family system: general family functioning (henceforth family functioning), problem solving, communication, roles, affective responsiveness, affective involvement, and behavior control. The family functioning subscale (12 items) was selected for the present study, as this subscale is widely used as a global index of overall family [37]. An example item is â In times of crisis we can turn to each other for supportâ . Raw scores were averaged; higher scores represent higher quality of family functioning. Cronbach Î± coefficient was .82, indicating good internal consistency.

**Peer connectedness.** Assessed during the lockdown via child-report with the *Adolescent Social Connection and Coping during COVID-19 Questionnaire* (ASC; [38]). This questionnaire was developed during the COVID-19 outbreak to assess adolescentsâ connection means and perceived connectedness when following physical distancing restrictions. Here, we only included the 4 items assessing connectedness to peers (friends whom they meet in person, friends whom they do not meet in person, online social networks and acquaintances of the same age). After asking how often adolescents connected by these means with their peers, we asked how socially connected this made them feel, on a scale from 1 = very socially disconnected to 7 = very socially connected. We computed a peer connectedness scale by averaging the 4 items; higher scores indicate higher peer connectedness. Internal consistency of this scale yielded a Cronbach Î± of .76.

## Statistical analyses

Logarithmic transformations were performed to the non-normally distributed dependent variable, improving its distribution. Second, the data was inspected for outliers and the values 3SD above and below the mean (two for internalizing problems and two for family functioning) were winsorized. Next, descriptive and correlation analyses were performed for all the study variables.

To answer the research questions, we ran a hierarchical regression model, with depressive symptoms as dependent variable. In the first step, sex, internalizing, externalizing symptoms, general family functioning, and peer connectedness were added. In the second step, to investigate buffering effects of family and peers, we added the interaction terms. To this end, the predictors were first mean-centered and then four interaction terms were computed by multiplying each predictor (internalizing, externalizing symptoms) by each moderator (family functioning, peer connectedness). The Q-Q plot of the residuals of this hierarchical model showed that they were sufficiently normally distributed.

# Results

## Descriptive analyses

In 2019, internalizing and externalizing symptoms, as assessed with the *Strengths and Difficulties Questionnaire*â were 10% and 16% above the clinical cut-off, respectively. During the lockdown, depressive symptoms, as assessed with the *The Center for Epidemiological Studies Depression Scale for Children*, were 24% above the cut-off (Table 1). Three adolescents had both pre-existing internalizing and externalizing symptoms above the clinical cut-off. Fifteen percent of adolescents with depressive symptoms during the lockdown had prior clinically significant internalizing symptoms, 11% externalizing symptoms, and only one had both above the clinical cut-off. Chi-square analyses showed non-significant sex differences on clinically relevant internalizing, externalizing and depressive symptoms (all $p > .281$). However, correlation analyses indicated that at a continuous level, sex was significantly associated with internalizing symptoms ($r = -.23$, $p = .015$) and depressive ($r = -.21$, $p = .024$) symptoms, suggesting that girls have higher scores of internalizing [$t(107) = 2.48$, $p = .015$] and depressive symptoms [$t(114) = 2.29$, $p = .024$] compared to boys. A modest positive correlation also emerged between internalizing and externalizing symptoms ($r = .31$, $p = .001$), suggesting that adolescents with higher internalizing symptoms also show higher externalizing symptoms. Mean

**Table 1. Descriptives and correlations amongst the variables in the study (N between 115â 109).**

|  | M (SD) | Range | % above clinical cut-off | 2. | 3. | 4. | 5. | 6. | 7. |
|---|---|---|---|---|---|---|---|---|---|
| **1. Age** | 13.06 (.56) | 12â 14 |  | .07 | -.08 | -.12 | .00 | .19* | .08 |
| **2. Sex (girls)** |  |  | 44(F) | - | **-.21*** | **-.23*** | -.12 | .06 | .06 |
| **3. Depressive symptoms** | 12.06 (7.05) | 2â 40 | 24.3 |  | - | **.25**** | .09 | **-.23*** | -.06 |
| **4. Internalizing symptoms** | 3.32 (2.75) | 0â 12.1 | 10.4 |  |  | - | **.31**** | .01 | .01 |
| **5. Externalizing symptoms** | 5.16 (2.91) | 0â 13 | 16.5 |  |  |  | - | .14 | **.25**** |
| **6. Family functioning** | 3.42 (.35) | 2.37â 4 |  |  |  |  |  | - | .15 |
| **7. Peer connectedness** | 3.34 (1.19) | 0â 5.25 |  |  |  |  |  |  | - |

*Note*. Sex: 0 = girls, 1 = boys; *M* = mean, *SD* = standard deviation.

* $p < .05$

** $p < .001$.

scores of internalizing and externalizing, and depressive symptoms were comparable to other Dutch adolescentsâ self-reports previous to the COVID-19 crisis [39].

## Main analyses

A regression analysis investigated whether earlier scores of internalizing and externalizing symptoms predict depressive symptoms scores during the first Dutch COVID-19 lockdown, and whether these relations are moderated by family functioning and peer connectedness. A summary of the results is provided in Table 2. The omnibus test for the first step with the main effects was significant [$R^2$ = .15, $F(5,99)$ = 3.44, $p$ = .007], while the second step with the interaction terms was non-significant [$ΔR^2$ = .02, $F(4,95)$ = 2.05, $p$ = .795]. Our results revealed significant main effects of internalizing symptoms [$β$ = 0.02, $t(104)$ = 2.21, $p$ = .029] and family functioning [$β$ = -0.18, $t(104)$ = -2.56, $p$ = .012]. These main effects indicate that higher internalizing symptoms one year earlier were associated with increased adolescent depressive symptoms, while better family functioning was associated with lower depressive symptoms during the COVID-19 lockdown. Depressive symptoms were not associated with externalizing symptoms or peer connectedness, nor were any of the interaction effects significant.

## Exploratory analyses

Next, sensitivity analyses with hierarchical regressions were conducted to tease apart specific associations for the internalizing symptoms subscales (i.e., emotional problems and peer problems). First, the variables were controlled for outliers: one value above 3SD for emotional problems was winsorized. The model with emotional problems mirrored the main analyses [$R^2$ = .16, $F(5,99)$ = 3.93, $p$ = .003], revealing a main effect of emotional problems [$β$ = 0.03, $t(104)$ = 2.66, $p$ = .009] and family functioning [$β$ = -0.18, $t(104)$ = -2.54, $p$ = .013] on depressive symptoms. In contrast, the model with peer problems [$R^2$ = .11, $F(5,99)$ = 2.44, $p$ = .039] revealed a non-significant main effect of peer problems [$β$ = 0.01, $t(104)$ = .66, $p$ = .507], and a main effect of family functioning only [$β$ = -0.18, $t(104)$ = -2.53, $p$ = .013] on depressive symptoms.

## Discussion

In the present longitudinal study with a low-risk community sample of young adolescents and their families, the results indicated that 24% of the adolescents experienced depressive

**Table 2. Moderated regression analysis with depressive symptoms as outcome variable ($N$ = 109).**

| Main effects | b | SE | β | t | p | CI |
|---|---|---|---|---|---|---|
| Constant | 1.59 | .24 | | 6.60 | .000 | 1.11â 2.07 |
| Sex | -.06 | .05 | -.12 | -1.25 | .215 | -.16 - .04 |
| Internalizing | .02 | .01 | .22 | 2.21 | **.029** | .00 - .04 |
| Externalizing | .00 | .01 | .05 | .51 | .609 | -.01 - .02 |
| Family functioning | -.18 | .07 | -.24 | -2.56 | **.012** | -.32 - -.04 |
| Peer connectedness | -.01 | .02 | -.02 | -.22 | .826 | -.05 - .04 |
| **2-way interactions** | | | | | | |
| Constant | 1.56 | .25 | | -.96 | .341 | -.15 â .05 |
| Internalizing*family functioning | -.00 | .03 | -.01 | -.13 | .894 | -.06 - .05 |
| Internalizing*peer connectedness | .00 | .01 | .07 | .65 | .520 | -.01 - .02 |
| Externalizing*family functioning | -.02 | .03 | -.09 | -.92 | .361 | -.07 - .03 |
| Externalizing*peer connectedness | -.00 | .01 | -.03 | -.29 | .768 | -.02 - .01 |

*Note.* $b$ = unstandardized coefficient, $SE$ = standard error, $β$ = standardized coefficient, $t$ = t-test value, $p$ value significance level at .05, $CI$ = confidence interval.

symptoms above the cut-off. Furthermore, we found that adolescents with pre-existing higher internalizing, but not externalizing, symptoms had higher depressive symptoms during the first COVID-19 lockdown. Moreover, better concurrent family functioning, but not peer connectedness, was linked to fewer depressive symptoms, irrespective of earlier internalizing symptoms.

One-in-four adolescents experienced clinically relevant depressive symptoms after 1â2 months of stringent COVID-19 lockdown, similar to other reports during the COVID-19 crisis [7]. Most recent meta-analytic data shows that rates of depression in youth worldwide have doubled during the pandemic compared to the prepandemic rates, reaching 25% prevalence of clinically relevant depressive symptoms [39]. The prevalence rates for adolescent clinical depression was previously estimated as 5.6% worldwide and 2.8% in The Netherlands [40], and up to 21â22% for subclinical symptoms [41,42]. The rates differ across ages, instruments and the clinical cut-offs used. What is more, Racine and colleagues [39] showed that mental health prevalence rates increased as the pandemic progressed, possibly suggesting that our findings from the beginning of the pandemic may have increased in the subsequent months. Yet, this remains to be determined in future longitudinal studies. Insofar, our results seem to indicate a worsening of young adolescentsâ mental health from 2019 to the period of heighted stress during the COVID-19 April/May lockdown. Although we only assessed internalizing symptoms, and not depressive symptoms, prior to the lockdown, 10% of our sample reported clinically meaningful internalizing symptomsâ assessed with the *Strengths and Difficulties Questionnaire*-, while 24% reported depressive symptomsâ assessed with *The Center for Epidemiological Studies Depression Scale for Children*- above the clinical cut-off a year later during the lockdown. Apparently, even in low-risk samples, a substantial group of adolescents and their families are vulnerable during times of crisis. It could be that adolescents from highly educated low-risk families such as those of the current study may be even less prepared to face a challenging crisis than adolescents who have previously experienced hardship. While high SES has previously been found to be a buffer against stress in children between 6â10 years [43], it may also be the case that children who do not normally experience disruptions in their daily environment are hit harder by an event or crisis such as the COVID-19 lockdown and its sudden restrictions. Indeed, some recent work found that low SES women were more resilient to stress compared to high SES women during the COVID-19 outbreak [44,45]. Despite the fact that the explanations for our findings are as yet unclear, the results clearly show that it is crucial to spread awareness about the most indicative signs of depression in adolescents, for parents and the surrounding community (e.g., teachers, sport instructors) to pay attention to (e.g., lack of appetite, sleep problems, general tiredness, being more quiet than usual, lack of energy to get started on activities, sadness). Identifying these signs early on may facilitate and accelerate timely interventions to support adolescentsâ mental wellbeing.

Furthermore, adolescents with higher internalizing symptoms in the year preceding the first COVID-19 wave were at heightened risk of having more depressive symptoms during this major crisis. Specifically, emotional problems, but not peer problems contributed to more depressive symptoms. This corroborates earlier findings that emotional challenges, such as poorer emotional regulation skills, influence childrenâs wellbeing and may exacerbate into later psychopathology [46]. In contrast, having externalizing symptoms did not predict depressive symptoms. Taken together, these findings tap into the debate of a homotypic (i.e., continuity of symptoms: internalizing symptoms predict later depressive symptoms) or a heterotypic trajectory of psychopathology (i.e., symptoms change throughout the lifespan: externalizing symptoms lead to later depressive symptoms). Our findings suggest a homotypic trajectory showing that internalizing symptoms are associated to later depressive symptoms, in line with previous findings [47]. While this continuity of internalizing symptoms into

depressive symptoms may not be unexpected, especially in times of crisis, what is remarkable is that this occurs even in low-risk samples, leaving open questions about risk and protective factors in the developmental trajectories of psychopathology. Alternatively, it still could be the case that externalizing symptoms might predict later depressive symptoms, following a hetero-typic pathway as found elsewhere [48], but that we were not able to capture this here possibly due to our limited sample size and one-year only timeline.

A key finding of our study is the positive role of the family environment for adolescentsâ mental health during the lockdown, irrespective of earlier symptoms. The family environment was related to fewer adolescentsâ depressive symptoms in times of crisis not only in already vulnerable children but in all. This evidence is however correlational and the associations could be bi-directional, opening the possibility that parenting a child with depressive symptoms may be more difficult and hence affect the general family functioning. During a period of high stress amid an unprecedented lockdown to contain COVID-19 infections, parents and their children were confined to their homes, needing to find a new structure in their daily lives, work, and schooling, while also having to deal with uncertainty and possibly emotional challenges. In such a situation, the general functioning of the family system appears to be highly important. This finding holds implications for the better and for the worse: while for some adolescents the family nucleus may act as a safe haven, for others it may be a stressor, as some reports on violence and abuse during the COVID-19 crisis have shown [49]. Noteworthy, in this study we investigated general family functioning. It is possible that specific family-related aspects play a more or less prominent role for adolescentsâ mental health. For example, our family functioning instrument includes subdomains such as communication, roles, affective responsiveness, and affective involvement. In a post-hoc exploratory analysis, we found indications that particularly role definition, affective responsiveness and affective involvement were associated with fewer depressive symptoms in the adolescents. Future studies with larger sample sizes and sufficient power should take a closer look at the role of these specific family-related aspects.

In addition to better family functioning, peer social connectedness was predicted to also contribute to reducing symptoms of depression, but this was unexpectedly not supported by the data. Social life was harshly disrupted during the COVID-19 lockdown, leading to physical isolation and fewer in-person encounters. However, we had expected that the adolescents would find some relief in online interactions or leisure activities, thanks to availability of social media, resulting in feelings of connectedness with their peers and reduced symptoms of depression. The fact that this was not the case may be because different aspects of socialization (e.g., physical isolation, digital socialization, physical or digital connectedness) may have dif-ferential impacts on adolescentsâ mental health. This needs to be studied further and in more depth.

A strength of our study is the longitudinal design, with two self-report assessments of men-tal wellbeing, and parent-report family assessment, at the peak of the first Dutch COVID-19 lockdown. This study uncovered that even in a low-risk community sample, one-in-four ado-lescents show clinically relevant symptoms of depression. While these results may not be immediately generalizable to other groups, the risk for mental health problems could be even higher for adolescents from harsher environments. A counterargument could be that adoles-cents who have experienced hardship before may show more resilience during crises like the COVID-19 pandemic. This remains to be determined in future work. Some limitations of this study are the limited sample size and the use of a non-validated questionnaire for peer con-nectedness, developed to capture the extraordinary lockdown circumstances. Also, our assess-ments of behavior were not identical at both waves. The aim of the longitudinal BIBO study was to examine early predictors of child behavioral development. For this reason, and also

because depressive symptoms are less common in early/middle childhood, we examined internalizing and externalizing behavior problems as broader, overarching constructs of behavior at age 12. However, when the COVID-19 pandemic started, the children were reaching adolescence, known for its increased vulnerability for depression [30]. Also, the first lockdown was characterized by high stress and anxiety due to a lack of knowledge about the virus and high mortality rates. For these reasons, we decided to specifically focus on depressive symptoms. However, to increase comparability, we recommend using similar assessments over age in longitudinal studies such as ours. Finally, the findings are limited to the first months of the lockdown. This highlights the need to pursue this investigation and characterize whether and how social and family dynamics, and their potential associations with adolescent mental health, changed as the COVID-19 pandemic progressed.

In conclusion, this study found that 24% young adolescents from a low-risk community sample showed clinically meaningful depressive symptoms during a stressful lockdown period. The risk for depressive symptoms was higher in adolescents with earlier internalizing symptoms, and with poor family functioning. Parents, teachers, and others in contact with adolescents should be made aware of the risk for serious mental health problems that may also affect adolescents from low-risk environments. In addition to paying attention to adolescentsâ warning signs and supporting them, it may be equally important to support the family system as a whole. This may indirectly protect the adolescents, and potentially other family members, from a deterioration of their mental wellbeing.

## Acknowledgments

We thank the children and parents who participated in the BIBO study as well PhD-students, research assistants and students for helping with data collection.

## Author Contributions

**Conceptualization:** Stefania V. Vacaru, Roseriet Beijers, Carolina de Weerth.

**Data curation:** Stefania V. Vacaru, Roseriet Beijers, Carolina de Weerth.

**Formal analysis:** Stefania V. Vacaru.

**Funding acquisition:** Roseriet Beijers, Carolina de Weerth.

**Investigation:** Roseriet Beijers, Carolina de Weerth.

**Project administration:** Roseriet Beijers, Carolina de Weerth.

**Resources:** Roseriet Beijers, Carolina de Weerth.

**Supervision:** Roseriet Beijers, Carolina de Weerth.

**Visualization:** Stefania V. Vacaru.

**Writing â original draft:** Stefania V. Vacaru, Roseriet Beijers, Carolina de Weerth.

**Writing â review & editing:** Stefania V. Vacaru, Roseriet Beijers, Carolina de Weerth.

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
