## [Decision Letter · Decision Letter 0]

10 Dec 2021

PONE-D-21-27377Internalizing Symptoms and Family Functioning Predict Adolescent Depressive Symptoms during COVID-19: a Longitudinal Study in a Community SamplePLOS ONE

Dear Dr. Vacaru,

Thank you for submitting your manuscript to PLOS ONE. After careful consideration, we feel that it has merit but does not fully meet PLOS ONE’s publication criteria as it currently stands. Therefore, we invite you to submit a revised version of the manuscript that addresses the points raised during the review process.

We look forward to receiving your revised manuscript.

Kind regards,

Anneloes van Baar, PhD

Academic Editor

PLOS ONE

“The BIBO study was supported by a Netherlands Organization for Scientific Research VIDI grant (575-25-009, to CdW), VICI grant (016.185.038, to CdW), VENI grant (016.195.197, to RB), an Early Career Award of the Royal Netherlands Academy of Arts and Sciences (to RB). The funders had no role in study design, data collection and analysis, decision to publish, or preparation of the manuscript.”

“The BIBO study was supported by a Netherlands Organization for Scientific Research VIDI grant (575-25-009, to CdW), VICI grant (016.185.038, to CdW), VENI grant (016.195.197, to RB), an Early Career Award of the Royal Netherlands Academy of Arts and Sciences (to RB). The funders had no role in study design, data collection and analysis, decision to publish, or preparation of the manuscript.”

Additional Editor Comments:

The study presents important information as it is able to relate pre-existing information regarding functioning and development of the adolescents to new information gathered during the first lockdown in The Netherlands. Both risk as well as protective factors could be used, specifically regarding overall family functioning and social connectedness.

Some clarification is needed in the manuscript.

In the Methods it is stated that data were collected in two waves: please explain when exactly.

In the Results the following is stated:

In 2019, internalizing and externalizing symptoms were 10 and 16% above the clinical cut-off,

191 respectively, whereas during the lockdown, depressive symptoms were 24% above the cut-off

192 (Table 1).

Please add the % of internalizing and externalizing symptoms above the clinical cut-off in 2020, as well as the % that showed depressive symptoms above clinical cut-off in 2019 – to allow more precise comparisons.

In the Discussion is stated:

Although we only assessed internalizing symptoms, and not depressive symptoms, prior

to the lockdown….

Why didn’t you study the specific relationship between prior depressive symptoms and depressive symptoms during the lockdown? Please explain either in the Discussion or clarify your research questions in this regard in the Introduction.

You also state:

It could be that adolescents from high educated, wealthy and low risk families may be even less prepared to face a challenging crisis than adolescents who have previously experienced hardship. How does this idea relate to your sample? Please explain.

Reviewers' comments:

Reviewer's Responses to Questions

**Comments to the Author**

1. Is the manuscript technically sound, and do the data support the conclusions?

Reviewer #1: Yes

2. Has the statistical analysis been performed appropriately and rigorously? 

Reviewer #1: Yes

3. Have the authors made all data underlying the findings in their manuscript fully available?

Reviewer #1: Yes

4. Is the manuscript presented in an intelligible fashion and written in standard English?

Reviewer #1: Yes

5. Review Comments to the Author

Reviewer #1: Reviewer’s comments

Thank you for the opportunity to review this paper. This is an interesting study, with clearly defined aim, generally well-written, nice longitudinal and multi-informant design, and with interesting results. I am positive about this study being published. Below, please find my comments, which I believe might improve the manuscript.

Introduction:

1. I think the first paragraph is too long. Could it be split in two more homogeneous paragraphs? Or maybe just cut shorter?

2. Line 63: I would say that life events are not “entered”, but experienced.

3. Lines 75-77: the aim can be written a bit more clearly: will you be looking at changes in symptoms pre- to during the first lockdown, or to moderation by pre-existing symptoms? The two previous sentences (lines 72-74) imply testing mean level changes, as they offer contrasting extant evidence for increase or decrease in symptoms. Please clarify the aim.

4. Lines 78-94: I think this paragraph is a bit confusing, as ideas regarding parenting and parent-adolescent relationship quality are intertwined and likely used interchangeably with family systems ideas. I think these two (parenting/parent-adolescent relationship quality and family functioning) are quite distinct concepts. Indeed, in my view family systems theories actually argue for this exact topic: how the system functions as a whole is not the same as how individuals behave (parenting) or how sub-systems within the system (e.g., the parent-adolescent dyad) interact. Please clarify.

5. Related to the above: If your family functioning measure has separate dimensions taping onto family as a system *and* parenting or p-a relationships more specifically, then it would be interesting to see what works best as a buffer: is it the system as a whole, or the parenting more specifically that might buffer during crises like the COVID-19?

6. Lines 90-94: please consider reporting contrasting or clarifying evidence for the potential effects (or absence thereof) of family functioning on adolescent mental health, e.g. Mastrotheodoros et al., 2020 https://doi.org/10.1007/s10964-019-01094-z

7. Lines 116-118: why would you hypothesize that adolescents with higher pre-existing externalizing would show higher internalizing? This comes a bit as a surprise, given the preceding introduction. A bit more clarity here would be welcome. Consider taking into account literature on the general psychopathology (p) factor for the possible interrelations between internalizing and externalizing.

Method:

8. Line 127: I get confused by the mean age reported here, compared to the one reported on line 124. Please clarify.

9. Lines 163-164: it looks like the citation (nr. 36) is wrong? I understand that this is a scale developed recently, but the reference is from 1997?

Results:

10. Lines 190-191: maybe good to remind the reader that those percentages come from different scales?

11. Lines 197-198: girls have higher scores, but the r correlations are negative. This is slightly confusing, not only because of the sign (a “higher” is indicated by a negative sign), but also because of the test statistic. Why r and not t-test? Maybe I am missing something here.

Discussion:

12. Lines 246-248: maybe good to remind the reader that different measures were applied pre- and during the pandemic.

13. Line 267: “across the lifespan” is a bit far-stretched, as there were only data from ages 13-14 in this study.

14. Lines 277-285: I think the evidence from the study suggested above (Mastrotheodoros et al., 2020) is also important here. In that study we found no evidence that family functioning has within-person effects on adolescent psychological symptoms, neither was there evidence for child mental health effects on family functioning. It might be that family functioning and adolescent mental health are only related on the between-person level. Given the results of that study, it would be better to avoid using causal language (line 278: “alleviate”), and keep the interpretation of your results on the between-person level.

6. PLOS authors have the option to publish the peer review history of their article (what does this mean?). If published, this will include your full peer review and any attached files.

Reviewer #1: **Yes: **Stefanos Mastrotheodoros

---

## [Author Response · Author response to Decision Letter 0]

17 Jan 2022

Prof. Anneloes van Baar

Editor

PLOSONE

17th January 2022

Dear Prof. van Baar,

We thank you for the opportunity to resubmit a revised version of our manuscript “Internalizing Symptoms and Family Functioning Predict Adolescent Depressive Symptoms during COVID-19: a Longitudinal Study in a Community Sample” for publication as an original paper in PLOSONE. 

We have addressed the comments provided by the reviewer and yourself, and indicated how this was done in a point-by-point fashion below. Based on these constructive and greatly appreciated comments, we improved the clarity of the manuscript with respect to the sample and methodology. Moreover, we now also discuss our findings in line with the interpersonal therapy framework, which highlights the clinical implications of our study. Additionally, we have improved the readability of the paper by adding some linguistic edits. 

This work is neither under consideration nor published elsewhere. All authors have approved the final submission of this manuscript and were meaningfully involved in the study. 

Overall, we believe that our manuscript has benefitted from the comments and as such is of interest to the readers of PLOSONE. We thank you for your further consideration.

Sincerely,

Stefania V. Vacaru, PhD., 

Roseriet Beijers, PhD., 

Carolina de Weerth, PhD. Prof.

Editor Comments:

1. In the Methods it is stated that data were collected in two waves: please explain when exactly.

 We clarified this on p.6 lines 265-266.

2. In the Results the following is stated: 

In 2019, internalizing and externalizing symptoms were 10 and 16% above the clinical cut-off,

respectively, whereas during the lockdown, depressive symptoms were 24% above the cut-off

(Table 1). Please add the % of internalizing and externalizing symptoms above the clinical cut-off in 2020, as well as the % that showed depressive symptoms above clinical cut-off in 2019 – to allow more precise comparisons.

In this study, we assessed internalizing/externalizing symptoms prior to the pandemic and depression symptoms during the pandemic. Therefore, the percentages of internalizing/externalizing during the pandemic, and the percentage of depressive symptoms prior to the pandemic are not available.

3. In the Discussion is stated:

Although we only assessed internalizing symptoms, and not depressive symptoms, prior

to the lockdown…. 

Why didn’t you study the specific relationship between prior depressive symptoms and depressive symptoms during the lockdown? Please explain either in the Discussion or clarify your research questions in this regard in the Introduction.

This study stems from a longitudinal study that started in late pregnancy and followed children across development (BIBO study). The aim of the BIBO study was to examine early predictors of child behavioral development. For this reason, and also because depressive symptoms are less common in early/middle childhood, we examined internalizing and externalizing behavior problems as more broader, overarching constructs of behavior. However, amidst the COVID-19 outbreak, we were worried about deterioration of mental health, especially also as our children had reached early adolescence, an age in which more children start to develop depressive symptoms. That our assessments of behavior were not identical at both waves is now included as a limitation to the Discussion section (pp. 14-15, lines 480-489).

4. You also state:

It could be that adolescents from high educated, wealthy and low risk families may be even less prepared to face a challenging crisis than adolescents who have previously experienced hardship. How does this idea relate to your sample? Please explain.

 We added an explanation of this idea to the Discussion (pp. 11-12, lines 396-403). We additionally took out the word ‘wealthy’ to characterize the families as this was an assumption not based on actual financial data.

Reviewer #1: 

1. I think the first paragraph is too long. Could it be split in two more homogeneous paragraphs? Or maybe just cut shorter?

 We have shortened the first paragraph accordingly (p. 3).

2. Line 63: I would say that life events are not “entered”, but experienced.

 We changed the wording as suggested (p. 3, line 59).

3. Lines 75-77: the aim can be written a bit more clearly: will you be looking at changes in symptoms pre- to during the first lockdown, or to moderation by pre-existing symptoms? The two previous sentences (lines 72-74) imply testing mean level changes, as they offer contrasting extant evidence for increase or decrease in symptoms. Please clarify the aim.

We have now clarified the aim (p. 4, lines 82-83).

4. Lines 78-94: I think this paragraph is a bit confusing, as ideas regarding parenting and parent-adolescent relationship quality are intertwined and likely used interchangeably with family systems ideas. I think these two (parenting/parent-adolescent relationship quality and family functioning) are quite distinct concepts. Indeed, in my view family systems theories actually argue for this exact topic: how the system functions as a whole is not the same as how individuals behave (parenting) or how sub-systems within the system (e.g., the parent-adolescent dyad) interact. Please clarify.

We thank the reviewer for pointing this out and have rewritten this paragraph to highlight the distinction between parenting/parent-adolescent relationships aspects, and family functioning (p. 4, lines 87-98).

5. Related to the above: If your family functioning measure has separate dimensions taping onto family as a system *and* parenting or p-a relationships more specifically, then it would be interesting to see what works best as a buffer: is it the system as a whole, or the parenting more specifically that might buffer during crises like the COVID-19?

This is a very interesting question. Given the rationale of the study to look at family functioning in general, and not specific sub-components, in combination with the lack of sufficient power, we did not run additional analyses with the subdomains of family-related aspects. We added these research questions as venues for future research to the discussion section, in combination with the outcomes of our exploratory analyses. The results of these analyses revealed significant main effects of roles, affective responsiveness and affective involvement (p. 13, lines 441-448). 

6. Lines 90-94: please consider reporting contrasting or clarifying evidence for the potential effects (or absence thereof) of family functioning on adolescent mental health, e.g. Mastrotheodoros et al., 2020 https://doi.org/10.1007/s10964-019-01094-z

We thank you for pointing us to this work, which we have now added to our introduction (p. 4, lines 95-96).

7. Lines 116-118: why would you hypothesize that adolescents with higher pre-existing externalizing would show higher internalizing? This comes a bit as a surprise, given the preceding introduction. A bit more clarity here would be welcome. Consider taking into account literature on the general psychopathology (p) factor for the possible interrelations between internalizing and externalizing.

We now clarify in our hypothesis that we expect that either internalizing or externalizing symptoms may be linked to higher depressive symptoms during the pandemic, in line with previous reports showing a positive relation between internalizing/externalizing symptoms and depression symptoms (e.g. Vinnakota & Kaur, 2018) (pp. 65-, lines 249-259).

8. Line 127: I get confused by the mean age reported here, compared to the one reported on line 124. Please clarify.

 We clarified the mean age and sd for the children at each assessment point (p. 6, lines 265-266).

9. Lines 163-164: it looks like the citation (nr. 36) is wrong? I understand that this is a scale developed recently, but the reference is from 1997?

 We corrected this accordingly on p. 8, line 305.

10. Lines 190-191: maybe good to remind the reader that those percentages come from different scales?

We highlighted now that the percentages reported stem from two different scales (p. 9, lines 331-334).

11. Lines 197-198: girls have higher scores, but the r correlations are negative. This is slightly confusing, not only because of the sign (a “higher” is indicated by a negative sign), but also because of the test statistic. Why r and not t-test? Maybe I am missing something here.

 We clarified the statistical test used to quantify the relation between the sex variable and internalizing or depressive symptoms and its interpretation in the descriptive analyses section. The sign depends on how sex was coded (0 or 1) and r refers to the correlation coefficient between the sex variable and internalizing or depressive symptoms. For clarity, we also added the t-test values (p. 9, lines 339-342).

12. Lines 246-248: maybe good to remind the reader that different measures were applied pre- and during the pandemic.

We have added that these symptoms were assessed with different instruments (p.11, lines 391-393).

13. Line 267: “across the lifespan” is a bit far-stretched, as there were only data from ages 13-14 in this study.

 We deleted this phrasing, p. 12, line 419 and p. 13, line 425.

14. Lines 277-285: I think the evidence from the study suggested above (Mastrotheodoros et al., 2020) is also important here. In that study we found no evidence that family functioning has within-person effects on adolescent psychological symptoms, neither was there evidence for child mental health effects on family functioning. It might be that family functioning and adolescent mental health are only related on the between-person level. Given the results of that study, it would be better to avoid using causal language (line 278: “alleviate”), and keep the interpretation of your results on the between-person level.

We thank you for pointing this out to us. We’ve now rephrased causal language where indicated and throughout the manuscript.

We also add here the funding statement.

Funding: The BIBO study was supported by a Netherlands Organization for Scientific Research VIDI grant (575-25-009, to CdW), VICI grant (016.185.038, to CdW), VENI grant (016.195.197, to RB), an Early Career Award of the Royal Netherlands Academy of Arts and Sciences (to RB). The funders had no role in study design, data collection and analysis, decision to publish, or preparation of the manuscript.

---

## [Editor Report · Decision Letter 1]

21 Feb 2022

Internalizing Symptoms and Family Functioning Predict Adolescent Depressive Symptoms during COVID-19: a Longitudinal Study in a Community Sample

PONE-D-21-27377R1

Dear Dr. Vacaru,

We’re pleased to inform you that your manuscript has been judged scientifically suitable for publication and will be formally accepted for publication once it meets all outstanding technical requirements.

Kind regards,

Anneloes van Baar, PhD

Academic Editor

PLOS ONE

Additional Editor Comments (optional):

Thank you very much for your response to the comments of the reviewer and the editor. You now have clarified and improved your paper. Good luck with your future work.
---

## [Editor Report · Acceptance letter]

11 Mar 2022

PONE-D-21-27377R1 

Internalizing Symptoms and Family Functioning Predict Adolescent Depressive Symptoms during COVID-19: a Longitudinal Study in a Community Sample 

Dear Dr. Vacaru:

I'm pleased to inform you that your manuscript has been deemed suitable for publication in PLOS ONE. Congratulations! Your manuscript is now with our production department. 

Kind regards, 

on behalf of

Dr. Anneloes van Baar 

Academic Editor

PLOS ONE